# DASGRAD:
# DOUBLE ADAPTIVE STOCHASTIC GRADIENT

## ABSTRACT

Adaptive moment methods have been remarkably successful for optimization under the presence of high dimensional or sparse gradients, in parallel to this, adaptive sampling probabilities for SGD have allowed optimizers to improve convergence rates by prioritizing examples to learn efficiently. Numerous applications in the past have implicitly combined adaptive moment methods with adaptive probabilities yet the theoretical guarantees of such procedures have not been explored. We formalize double adaptive stochastic gradient methods DASGRAD as an optimization technique and analyze its convergence improvements in a stochastic convex optimization setting, we provide empirical validation of our findings with convex and non convex objectives. We observe that the benefits of the method increase with the model complexity and variability of the gradients, and we explore the resulting utility in extensions to transfer learning.

## 1 INTRODUCTION AND MOTIVATION

Stochastic gradient descent (SGD) is a widely used optimization method, and currently through backpropagation this algorithm has propelled the success of many deep learning applications. The Deep Learning community has particularly adopted variants of adaptive moment methods for SGD that specialize in high-dimensional features and non convex objectives, examples include ADAGRAD, ADADELTA, RMSPROP, ADAM and AMSGRAD (Duchi et al. (2011); Zeiler (2012); Tieleman & Hinton (2012); Kingma & Ba (2014); Reddi et al. (2018)). All these adaptive moment methods relied on the efficient use of the information of the geometry of the problem to improve the rate of convergence.

In parallel to the previous ideas, adaptive probabilities methods for SGD, traditionally focusing on convex objectives, have shown advantages over its uniform sampling baselines, by allowing a more efficient use of the gradient information (Zhu (2018); Shen et al. (2016); Bottou et al. (2016); Csiba et al. (2015); Stich et al. (2017)).

Adaptive probabilities were introduced to the Deep Learning community by Hinton (2007) through the discriminative fine tuning procedure. The method was further explored in a range of applications like object detection, reinforcement learning and curriculum learning (Shrivastava et al. (2016); Schaul et al. (2015); Bengio et al. (2009)). In these examples the combination of adaptive moments and adaptive probabilities methods was implicit and its analysis as a pure optimization technique was still an open question.

In this paper we analyze the asymptotic convergence properties of combining adaptive probabilities and adaptive moments. To our knowledge such family has not yet been introduced as an optimization procedure. We will refer to this family of optimization algorithms as Double Adaptive Stochastic Gradient DASGRAD.

We show that the improvements of DASGRAD depend on its flexibility to control the variance of the adaptive moments methods. We prove the theoretical guarantees and improvements in a convex setting and validate these observations empirically in convex and deep learning objectives. Finally we demonstrate the generalization properties of the algorithm with a simple extension to importance weight transfer learning.

## 2   ADAPTIVE GRADIENT METHODS

**Notation.** In order to facilitate the proofs and reading process we introduce some simplified notation that will be common to the analyzed algorithms. Let $\boldsymbol{a}, \boldsymbol{b} \in \mathbb{R}^d$ and $\boldsymbol{M} \in S_+^d$, then the multiplication of vector $\boldsymbol{a}$ by the inverse of $\boldsymbol{M}$ will be $\boldsymbol{M}^{-1}\boldsymbol{a} = \boldsymbol{a}/\boldsymbol{M}$. Let $\sqrt{\boldsymbol{a}}$ be the element-wise square root of vector $\boldsymbol{a}$, $\boldsymbol{a}^2$ the element-wise square, $\boldsymbol{a}/\boldsymbol{b}$ the element-wise division, and $\max(\boldsymbol{a}, \boldsymbol{b})$ the element-wise max of vector $\boldsymbol{a}$ and vector $\boldsymbol{b}$. Finally for any natural $n$ the set $\{1, \ldots, n\}$ is denoted as $[n]$.

Let $\mathcal{T} = \{(\mathbf{x}_i, \mathbf{y}_i)\}_{i=1}^n$ be a training set; let $f : \Theta \times X \times Y \to \mathbb{R}$ be a differentiable function that represents the empirical risk of an agent over $\mathcal{T}$ for the parameters $\boldsymbol{\theta} \in \Theta$, with $\Theta \subseteq \mathbb{R}^d$ a convex feasible set of parameters; let $S_+^d$ the set of positive definite matrices in $\mathbb{R}^{d \times d}$, for a given matrix $\boldsymbol{M} \in S_+^d$ and parameter $\boldsymbol{\theta}' \in \Theta$; let $\Pi_{\Theta, \boldsymbol{M}}$ be the projection operator over the feasible domain defined by $\Pi_{\Theta, \boldsymbol{M}}(\boldsymbol{\theta}') = \arg\min_{\boldsymbol{\theta} \in \Theta} ||\boldsymbol{M}^{1/2}(\boldsymbol{\theta} - \boldsymbol{\theta}')||$. [1]

For the iterative stochastic optimization algorithm $\mathcal{A}$, let $i_t$ be a sampled index $i$ at step $t$ drawn from the training set indices $[n]$, with $i_t \sim \boldsymbol{p}_t$ and $\boldsymbol{p}_t \in \Delta_+^n = \{\boldsymbol{p} \in \mathbb{R}^n : p_i > 0 \quad \Sigma_i p_i = 1\}$. We denote the evaluated risk $f(\boldsymbol{\theta}, \mathbf{x}_i, \mathbf{y}_i) = f_i(\boldsymbol{\theta})$, the complete gradient $\nabla f(\boldsymbol{\theta}_t) = \frac{1}{n}\Sigma_{i_t} \nabla f_{i_t}(\boldsymbol{\theta}_t)$ and the stochastic gradient $\nabla f_{i_t}(\boldsymbol{\theta}_t)$, analogous a full descent direction $\boldsymbol{m}_t = \frac{1}{n}\Sigma_{i_t} \boldsymbol{m}_{i_t}$ and a stochastic descent direction $\boldsymbol{m}_{i_t}$.

**Stochastic Optimization Framework.** To analyze the convergence of the stochastic optimization algorithm $\mathcal{A}$ we use the convex optimization setting where we assume that the objective function is convex with bounded gradients, that is $||\nabla f_i(\boldsymbol{\theta})||_\infty \leq G$ for all $i \in [n]$, $\boldsymbol{\theta} \in \Theta$, and finally the parameter space $\Theta$ has bounded diameter, that is $||\boldsymbol{\theta} - \boldsymbol{\theta}'||_\infty \leq D$ for all $\boldsymbol{\theta}, \boldsymbol{\theta}' \in \Theta$.

For our purposes, the algorithm $\mathcal{A}$ at time $t$ chooses a distribution over the training set $\hat{\boldsymbol{p}}_t \in \Delta_+^n$, obtains a training example $i_t \sim \hat{\boldsymbol{p}}_t$ and its importance weights $\hat{w}_{i_t} = (1/n)/p_{i_t}$, then updates its parameters $\boldsymbol{\theta}_t \in \Theta$ using the available data at time $t$ and the importance weights $\hat{w}_{i_t}$ to unbias the direction of the gradients. After the update, the algorithm incurs in a loss from an unknown function $f(\boldsymbol{\theta}_t)$. To assess the performance of the algorithm after $T$ steps we use the expected regret, which measures the difference of the loss at time $t$ and the loss for optimal fixed parameter, along the possible trajectories induced by the chosen probabilities.

$$R(\mathcal{A}) = \sum_{t=1}^T \mathbb{E}_n \left[ f_i(\boldsymbol{\theta}_t) - \min_{\boldsymbol{\theta}} \mathbb{E}_n[f_i(\boldsymbol{\theta})] \right]$$

The goal is to design an algorithm $\mathcal{A}$ that has sub linear expected regret $R(\mathcal{A})_T = \mathcal{O}(T)$, which in turn implies that the algorithm will converge on average to the optimal parameter.

---

**Algorithm 1: Double Adaptive Methods**

**Input:** $\boldsymbol{\theta}_1 \in \Theta$, step size $\{\alpha_t > 0\}_{t=1}^T$, functions $\{\phi_t, \psi_t\}_{t=1}^T$
**for** $t = 1$ **to** $T$ **do**
    Choose $\hat{\boldsymbol{p}}_t \in \Delta_+^n$, and sample $i_t \sim \hat{\boldsymbol{p}}_t$
    Calculate $\boldsymbol{g}_{i_t} = \nabla f_{i_t}(\boldsymbol{\theta}_t)$ and $\hat{w}_{i_t} = (1/n)/\hat{p}_{i_t}$
    $\boldsymbol{m}_{i_t} = \phi_t(\boldsymbol{g}_{i_1}, \ldots, \boldsymbol{g}_{i_t})$ and $\hat{\boldsymbol{V}}_{i_t} = \psi_t(\boldsymbol{g}_{i_1}, \ldots, \boldsymbol{g}_{i_t})$
    $\hat{\boldsymbol{\theta}}_{t+1} = \boldsymbol{\theta}_t - \alpha_t \hat{w}_{i_t} \hat{\boldsymbol{V}}_{i_t}^{-1/2} \boldsymbol{m}_{i_t}$
    $\boldsymbol{\theta}_{t+1} = \Pi_{\Theta, \hat{\boldsymbol{V}}_{i_t}^{1/2}}(\hat{\boldsymbol{\theta}}_{t+1})$

---

Algorithm 1 constitutes the general family of double adaptive gradient methods. This algorithm comprehends the classical stochastic gradient descent, adaptive moment methods family Zeiler (2012); Tieleman & Hinton (2012); Kingma & Ba (2014); Reddi et al. (2018), and second order methods Duchi et al. (2011), varying the averaging functions of the past gradients with $\phi_t : \Theta^t \to \mathbb{R}^d$, and approximating the Hessian matrix with the functions $\psi_t : \Theta^t \to S_+^d$.

---

[1] The projection operator enables the algorithm to deal with constrained optimization over compact convex domains that are equivalent to common regularization techniques like ridge and LASSO.

**Adaptive Probabilities Methods.** The classic stochastic gradient descent algorithm is recovered with the following step size, sampling probabilities and functions:

$$\alpha_t = \alpha/\sqrt{t} \quad p_{i_t} = 1/n \ \text{ for all } t \in [T], i \in [n]$$
$$\phi_t(\boldsymbol{g}_{i_1}, \ldots, \boldsymbol{g}_{i_t}) = \boldsymbol{g}_{i_t} \ \psi_t(\boldsymbol{g}_{i_1}, \ldots, \boldsymbol{g}_{i_t}) = \boldsymbol{I} \tag{SGD}$$

Adaptive probabilities methods can be obtained simply by allowing the algorithm to choose a different probability $\hat{\boldsymbol{p}}_t$ at any time $t$:

$$\alpha_t = \alpha/\sqrt{t} \quad \hat{\boldsymbol{p}}_t \in \Delta_+^n \ \text{ for all } t \in [T]$$
$$\phi_t(\boldsymbol{g}_{i_1}, \ldots, \boldsymbol{g}_{i_t}) = \boldsymbol{g}_{i_t} \ \psi_t(\boldsymbol{g}_{i_1}, \ldots, \boldsymbol{g}_{i_t}) = \boldsymbol{I} \tag{AP-SGD}$$

Significant improvements in the convergence rate of the algorithm can be obtained by cleverly choosing and computing such probabilities that in turn enables the algorithm to use data in a more efficient manner Stich et al. (2017). Fixed importance sampling is the case when $\hat{\boldsymbol{p}}_t = p$ for all $t \in [T]$.

**Adaptive Moments Methods.** Duchi et al. propelled interest and research on adaptive algorithms. In their work they noticed that SGD lacked good convergence behavior in sparse settings, and proposed a family of algorithms that allowed the methods to dynamically incorporate information about the geometry of the data Duchi et al. (2011). Following huge gains obtained with ADAGRAD, the deep learning community proposed variants based on exponential moving average functions for $\psi_t$ like ADADELTA, RMSPROP, ADAM and most recently AMSGRAD (Zeiler (2012); Tieleman & Hinton (2012); Kingma & Ba (2014); Reddi et al. (2018)).

The first algorithm ADAGRAD is obtained by the following proximal functions:

$$\alpha_t = 1/\sqrt{t} \quad p_{i_t} = 1/n \ \text{ for all } t \in [T], i \in [n]$$
$$\phi_t(\boldsymbol{g}_{i_1}, \ldots, \boldsymbol{g}_{i_t}) = \boldsymbol{g}_{i_t}$$
$$\psi_t(\boldsymbol{g}_{i_1}, \ldots, \boldsymbol{g}_{i_t}) = \frac{1}{t}\text{diag}(\Sigma_{\tau=1}^t \boldsymbol{g}_{i_\tau}^2) \tag{ADAGRAD}$$

The ADAM/AMSGRAD algorithm is obtained by setting:

$$\alpha_t = 1/\sqrt{t} \quad p_{i_t} = 1/n \ \forall t \in [T], i \in [n]$$
$$\phi_t(\boldsymbol{g}_{i_1}, \ldots, \boldsymbol{g}_{i_t}) = \Sigma_{\tau=1}^t \beta_1(t)_\tau \boldsymbol{g}_{i_\tau}$$
$$\boldsymbol{v}_{i_t} = (1 - \beta_2)\Sigma_{\tau=1}^t \beta_2^{t-\tau} \boldsymbol{g}_{i_\tau}^2 \quad \hat{\boldsymbol{v}}_{i_t} = \max(\hat{\boldsymbol{v}}_{i_{t-1}}, \boldsymbol{v}_{i_t}) \tag{ADAM/AMSGRAD}$$
$$\psi_t(\boldsymbol{g}_{i_1}, \ldots, \boldsymbol{g}_{i_t}) = \text{diag}(\hat{\boldsymbol{v}}_{i_t})$$

Fortunately a very simple and computationally efficient way to implement ADAM is given by a recursion. RMSPROP is the particular case of ADAM when $\beta_1 = 0$ and without maximum operator for the second moments vector, while ADAM is recovered without the maximum operator.

**Double Adaptive Methods.** The key idea behind both the adaptive probabilities methods and adaptive moment methods is the efficient use of the information available in the training data to improve the convergence of the algorithms. In the case of adaptive moment methods the diagonal approximations of the Hessian matrix use the information about the geometry of the problem while the adaptive sampling methods, the probabilities $\hat{p}_{i_t}$, prioritize the examples with the highest impact on the learning progress. As these improvements rely on complementary sources of information we can combine them into a general framework described by the double adaptive methods in Algorithm 1.

To analyze the theoretical improvement guarantees of the double adaptive methods we first we extend the adaptive moments convergence guarantees to the stochastic case with uniform sampling and then we compare them to the convergence guarantees using optimal probabilities.

## 3 CONVERGENCE ANALYSIS

### 3.1 CONVERGENCE OF ADAPTIVE MOMENTS METHODS

We first provide a regret bound of ADAM for weakly convex objectives with uniform probabilities adapting the arguments in Reddi et al. (2018) and Kingma & Ba (2014). Then we extend these results to the adaptive probabilities case.

**Theorem 1.** *Let $\{\boldsymbol{\theta}_t\}_{t=1}^{T}$ be the sequence obtained with* ADAM*, then the regret bound is:*

$$R(\text{ADAM}) \leq \sum_{t=1}^{T} \frac{1}{2\alpha_t(1-\beta_{1_t})} \mathbb{E}_n \left[ ||\hat{\boldsymbol{V}}_{i_t}^{1/4}(\boldsymbol{\theta}_t - \boldsymbol{\theta}^*)||^2 - ||\hat{\boldsymbol{V}}_{i_t}^{1/4}(\boldsymbol{\theta}_{t+1} - \boldsymbol{\theta}^*)||^2 \right]$$

$$+ \sum_{t=1}^{T} \frac{\alpha_t}{2(1-\beta_{1_t})} \mathbb{E}_n \left[ ||\hat{\boldsymbol{V}}_{i_t}^{-1/4} \boldsymbol{m}_{i_t}||^2 \right]$$

$$+ \sum_{t=1}^{T} \frac{\alpha_t \beta_{1_t}}{2(1-\beta_{1_t})} ||\hat{\boldsymbol{V}}_t^{-1/4} \boldsymbol{m}_{t-1}||^2 + \sum_{t=1}^{T} \frac{\beta_{1_t}}{2\alpha_t(1-\beta_{1_t})} ||\hat{\boldsymbol{V}}_t^{1/4}(\boldsymbol{\theta}_t - \boldsymbol{\theta}^*)||^2$$

**Corollary 1.1.** *Following the sequence $\{\boldsymbol{\theta}_t\}_{t=1}^{T}$ of* ADAM *with step size $\alpha_t = \alpha/\sqrt{t}$, averaging parameters $\beta_1 = \beta_{1_1}$, $\beta_{1_t} \leq \beta_1$ for all $t \in [T]$, $\gamma = \beta_1/\sqrt{\beta_2} < 1$ and uniform probabilities $p_{i_t} = 1/n$. If we assume that $\Theta$ has bounded diameter $D$, $||\nabla f_{i_t}(\boldsymbol{\theta})||_\infty \leq G$ for all $t \in [T]$ and $\boldsymbol{\theta} \in \Theta$, then the expected regret bound is:*

$$R(\text{ADAM}) \leq \frac{D^2\sqrt{T}}{2\alpha(1-\beta_1)} \mathbb{E}_n \left[ ||\hat{\boldsymbol{v}}_{i_T}^{1/4}||^2 \right]$$

$$+ \frac{\alpha\sqrt{1+\log(T)}}{2(1-\beta_1)^2\sqrt{(1-\beta_2)}(1-\gamma)} \sum_{h=1}^{d} || \,|\bar{\boldsymbol{g}}|_{1:T,h}\, ||$$

$$+ \frac{\alpha G d}{2\alpha(1-\beta_1)^3\sqrt{(1-\beta_2)}(1-\gamma)} + \frac{D^2}{2\alpha(1-\beta_1)} \sum_{t=1}^{T} \sqrt{t}\beta_1^{T-t} ||\hat{\boldsymbol{v}}_t^{1/4}||^2$$

### 3.2 CONVERGENCE OF DOUBLE ADAPTIVE METHODS

**Theorem 2.** *Let $\{\boldsymbol{\theta}_t\}_{t=1}^{T}$ be a* DASGRAD *sequence, for a trajectory $p_t \in \Delta_+^n$ the regret bound is:*

$$R(\text{DASGRAD}) \leq \sum_{t=1}^{T} \frac{1}{2\alpha_t(1-\beta_{1_t})} \mathbb{E}_{\boldsymbol{p}_{1:t}} \left[ ||\hat{\boldsymbol{V}}_{i_t}^{1/4}(\boldsymbol{\theta}_t - \boldsymbol{\theta}^*)||^2 - ||\hat{\boldsymbol{V}}_{i_t}^{1/4}(\boldsymbol{\theta}_{t+1} - \boldsymbol{\theta}^*)||^2 \right]$$

$$+ \sum_{t=1}^{T} \frac{\alpha_t}{2(1-\beta_{1_t})} \mathbb{E}_{\boldsymbol{p}_{1:t}} \left[ w_{i_t}^2 ||\hat{\boldsymbol{V}}_{i_t}^{-1/4} \boldsymbol{m}_{i_t}||^2 \right]$$

$$+ \sum_{t=1}^{T} \frac{\alpha_t \beta_{1_t}}{2(1-\beta_{1_t})} ||\hat{\boldsymbol{V}}_t^{-1/4} \boldsymbol{m}_{i_{t-1}}||^2 + \sum_{t=1}^{T} \frac{\beta_{1_t}}{2\alpha_t(1-\beta_{1_t})} ||\hat{\boldsymbol{V}}_t^{1/4}(\boldsymbol{\theta}_t - \boldsymbol{\theta}^*)||^2$$

**Corollary 2.1.** *Following the sequence $\{\boldsymbol{\theta}_t\}_{t=1}^{T}$ of* DASGRAD*, step size $\alpha_t = \alpha/\sqrt{t}$, averaging parameters $\beta_1 = \beta_{1_1}$, $\beta_{1_t} \leq \beta_1$ for all $t \in [T]$, $\gamma = \beta_1/\sqrt{\beta_2} < 1$ and the optimal adaptive probabilities $\hat{p}_{i_t} \propto ||\hat{\boldsymbol{V}}_{i_t}^{-1/4}\boldsymbol{m}_{i_t}||$. If we assume that $\Theta$ has bounded diameter $D$ and $||\nabla f_{i_t}(\boldsymbol{\theta})||_\infty \leq G$ for all $t \in [T]$ and $\boldsymbol{\theta} \in \Theta$, then the expected regret bound is:*

$$R(\text{DASGRAD}) \leq \frac{D^2\sqrt{T}}{2\alpha(1-\beta_1)} \mathbb{E}_{\hat{\boldsymbol{p}}_{1:T}} \left[ ||\hat{\boldsymbol{v}}_{i_T}^{1/4}||^2 \right]$$

$$+ \frac{\alpha\sqrt{1+\log(T)}}{2(1-\beta_1)^2\sqrt{(1-\beta_2)}(1-\gamma)} \sum_{h=1}^{d} || \,|\bar{\boldsymbol{g}}|_{1:T,h}\, || - \sum_{t=1}^{T} Var_n \left( ||\hat{\boldsymbol{V}}_{i_t}^{1/4}\boldsymbol{m}_{i_t}|| \right)$$

$$+ \frac{\alpha G d}{2\alpha(1-\beta_1)^3\sqrt{(1-\beta_2)}(1-\gamma)} + \frac{D^2}{2\alpha(1-\beta_1)} \sum_{t=1}^{T} \sqrt{t}\beta_1^{T-t} ||\hat{\boldsymbol{v}}_t^{1/4}||^2$$

### 3.3 CONVERGENCE COMPARISON

Adaptive moment methods can improve classical gradient descent by integrating the geometry of the problem with a diagonal approximation of the Hessian an may achieve an exponentially smaller bound for the expected regret with respect to the dimensions of the input data $d$, when dealing with sparse features or small gradients in general. As shown by Duchi et al. (2011) for the adaptive moment methods in the sparse setting, the potential component and error component of the expected regret of Corollary 1.1 each will satisfy:

$$\mathbb{E}_{\boldsymbol{p}_{1:T}}\left[||\hat{\boldsymbol{v}}_{i_T}^{1/4}||^2\right] = \mathbb{E}_{\boldsymbol{p}_{1:T}}\left[\sum_{h=1}^{d}\hat{\boldsymbol{v}}_{i_T,h}^{1/2}\right] \ll \sqrt{d} \quad \text{and} \quad \sum_{h=1}^{d}||\,|\bar{\boldsymbol{g}}\,|_{1:T,h}\,|| \ll \sqrt{dT}$$

which in turn translates to a much better expected regret bound than $\mathcal{O}(\sqrt{dT})$ for classic SGD on weakly convex objectives and sparse inputs.

In parallel to the previous improvements, the adaptive probabilities methods can further speed the convergence by allowing the algorithm to evaluate the relative importance of each data point to maximize the expected learning progress, and minimize the variance of the stochastic gradient at each step. Given the optimal adaptive probabilities the error component of the expected regret in Corollary 2.1 satisfies:

$$\sum_{h=1}^{d}||\,|\bar{\boldsymbol{g}}\,|_{1:T,h}\,|| - \sum_{t=1}^{T}\text{Var}_n\left(||\hat{\boldsymbol{V}}_{i_t}^{1/4}\boldsymbol{m}_{i_t}||\right) \ll \sum_{h=1}^{d}||\,|\bar{\boldsymbol{g}}\,|_{1:T,h}\,||$$

This shows how the optimal adaptive sampling probabilities on a convex setting can further improve the convergence rate by allowing to flexibly control the variance of the gradients.

## 4 DASGRAD IMPLEMENTATION

To obtain the optimal probabilities it is necessary to compute the norm of the gradient for each training sample at each step. Given the deep learning libraries today, this calculation renders optimal adaptive probabilities methods impractical for real applications [2]. Still, for completeness of the exposition of the theoretical results we provide empirical evidence of the convergence improvements, to do it we use an approximation of the optimal algorithm from the double adaptive methods, following Algorithm 2.

---

**Algorithm 2: DASGRAD approximation**

---

**Input:** $\boldsymbol{\theta}_1 \in \Theta$, functions $\{\phi_t, \psi_t\}_{t=1}^{T}$, frequency $J$
**for** $t = 1$ **to** $T$ **do**
    **if** $t \bmod J = 0$ **then**
        | Compute $\hat{\boldsymbol{p}}_t \in \Delta_+^n$ setting $\hat{\boldsymbol{p}}_{i_t} \propto ||\hat{\boldsymbol{V}}_{i_t}^{-1/4}\boldsymbol{m}_{i_t}|| + \epsilon$
    Sample $i_t \sim \hat{\boldsymbol{p}}_t$ using the segment tree
    Calculate $\boldsymbol{g}_{i_t} = \nabla f_{i_t}(\boldsymbol{\theta}_t)$ and $\hat{w}_{i_t} = (1/n)/\hat{p}_{i_t}$
    $\boldsymbol{m}_t = \beta_{1t}\boldsymbol{m}_{t-1} + (1 - \beta_{1t})\boldsymbol{g}_t$ and $\boldsymbol{v}_t = \beta_2\boldsymbol{v}_{t-1} + (1 - \beta_2)\boldsymbol{g}_t^2$
    $\hat{\boldsymbol{v}}_t = max(\hat{\boldsymbol{v}}_{t-1}, \boldsymbol{v}_t)$ and $\hat{\boldsymbol{V}}_t = diag(\hat{\boldsymbol{v}}_t)$
    $\hat{\boldsymbol{\theta}}_{t+1} = \boldsymbol{\theta}_t - \alpha_t\hat{w}_{i_t}\hat{\boldsymbol{V}}_{i_t}^{-1/2}\boldsymbol{m}_{i_t}$
    $\boldsymbol{\theta}_{t+1} = \Pi_{\Theta,\hat{\boldsymbol{V}}_{i_t}^{1/2}}(\hat{\boldsymbol{\theta}}_{t+1})$

---

[2]Current dynamical computational graph libraries compute the average gradient batches by default and available workarounds are still slow. See https://github.com/pytorch/pytorch/issues/7786 and https://github.com/tensorflow/tensorflow/issues/4897.

## 5 EMPIRICAL RESULTS

In this section we provide empirical evidence of the convergence rates on classification problems using logistic regression and deep neural networks, using ADAM, AMSGRAD, and DASGRAD.

**Logistic Regression**: For the convex setting we solve two classification problems with L2 regularization. For the non sparse feature experiment we use the MNIST digit dataset, which is composed of $60,000$ images of $28 \times 28$ hand written digits. For the sparse feature experiment we use the IMDB movie rating dataset which is composed of $25,000$ highly polar movie reviews and the sentiment label for the review Maas et al. (2011).[3]

**Neural Networks**: For the non convex setting we perform one experiment, we use the CIFAR10 dataset, which is composed of $60,000$ colour images of $32 \times 32$ pixels labeled in 10 classes. For this multiclass classification problem we use a convolutional neural network following the SMALL-CIFARNET architecture, consisting of two convolution filters combined with max pooling and local response normalization, followed by two fully connected layers of rectified linear units Krizhevsky et al. (2012). [4]

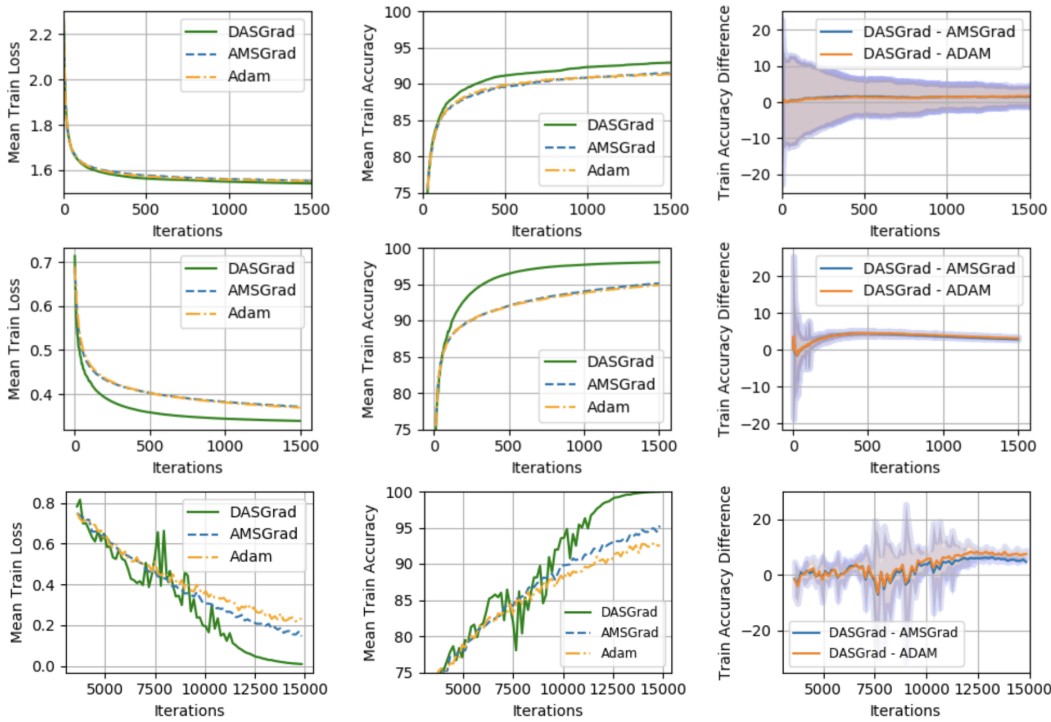

Figure 1: Trajectories in convex and deep learning settings. First row 100 logistic regressions on MNIST, second row 100 logistic regressions on IMDB, third row 10 convolutional neural networks on CIFAR10. We show the mean over the trajectories for training loss (left), training accuracy (center), and the accuracy improvement of DASGRAD with respect to AMSGRAD and ADAM with 95% confidence intervals (right).

---

[3]For both experiments, we use a batch of size 32, with a probability update every 10 steps, and the step size $\alpha_t = \alpha/\sqrt{t}$. We set $\beta_1 = 0.9$, $\beta_2 = 0.99$, and choose $\alpha$ through a grid search. For the MNIST dataset, for all three optimizers, the optimal learning rates are $\alpha = 0.01$. For the IMDB dataset, we find the optimal learning rates to be $\alpha = 0.005$ for ADAM, $\alpha = 0.006$ for AMSGRAD, and $\alpha = 0.02$ for DASGRAD.

[4]For the experiment we use a batch size of 32, with a probability update every 300 steps, and step size of $\alpha_t = \alpha/\sqrt{t}$. We set $\beta_1 = 0.9$, $\beta_2 = 0.99$, and choose $\alpha$ through a grid search, for which the optimal learning rate for all optimizers is $\alpha = 0.001$.

From the comparison in Figure 1, we observe that in all cases the DASGRAD optimization algorithm outperforms its adaptive moment counterparts represented by ADAM and AMSGRAD, as predicted by the theoretical analysis. The improvement is more significant for the IMDB dataset than it is for the MNIST dataset. From Figure 1 we can see that DASGRAD continues to outperform ADAM and AMSGRAD in the deep learning setting. These results reinforce the previous statement that the benefits from DASGrad increase with the complexity of the data and the models.

## 6 DISCUSSION

### 6.1 CONVERGENCE IMPROVEMENTS AND VARIANCE OF GRADIENTS

To further explore the relationship between variance and the improvements to the convergence rate of the DASGRAD algorithm, we implemented an online centroid learning experiment. Because of the linear relationship between the features and the gradients, we are able to explicitly control their variance. For this experiment, the empirical risk and gradients will be given by
$R_n(\boldsymbol{\theta}) = \frac{1}{2n} \sum_{i=1}^{n} ||\boldsymbol{\theta} - \mathbf{x}_i||^2$ and $\nabla f(\boldsymbol{\theta}, \mathbf{x}_i) = \boldsymbol{\theta} - \mathbf{x}_i$.

As we can see from Figure 2 the greater the variance of the gradients, the greater the benefit that one can obtain from an adaptive probabilities method such as DASGRAD in convex objectives, since those probabilities will prioritize the data points with the most learning potential.

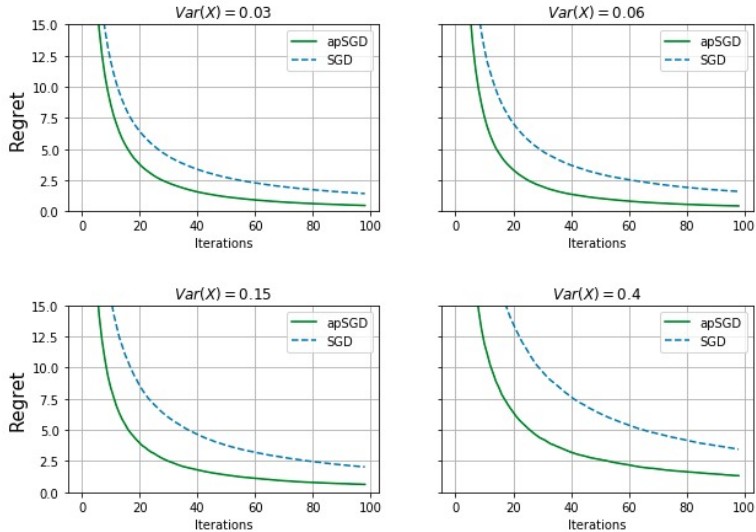

Figure 2: Trajectories of 100 random seeds, for the online centroid learning problem with different variance for the features. Enhanced improvements of adaptive methods with higher variance of the gradients.

### 6.2 FLEXIBLE CONTROL OF VARIANCE

Recent insights on the generalization properties of minibatch SGD for non convex objectives suggest that higher variance gradients tend to converge to flatter regions of the loss surfaces (Keskar et al. (2016)). Applications like curriculum learning that shape the learning procedure, by gradually making the task more difficult through importance sampling, may allow to maintain a higher variance of the gradients for longer steps, this combined with the previous intuitions offers an explanation of the mathematical basis of its success.

While curriculum learning is contrary to the optimal probabilities of the double adaptive methods for convex settings, the underlying principle of flexible control of the variance of the gradients operates as the mechanism behind both procedures. This observations strengthens the argument that improving our understanding of the implicit optimization techniques in this algorithms can also improve our understanding that so far has relied mostly on intuitive explanations of their success.

### 6.3 IMPORTANCE WEIGHT TRANSFER LEARNING

When the training $\mathcal{T}$ and test $\mathcal{T}'$ set do not share the same distribution, we may face a distribution mismatch problem. The DASGRAD algorithm is compatible with the cost re-weighting correction technique Elkan (2001); Bowyer et al. (2011) as we can set the importance weights $w_t$ for any trajectory of distributions $\boldsymbol{p}_t$, to unbias the gradients for the test distribution instead of the training.

$$R(\text{DASGRAD})_{\mathcal{T}'} = \sum_{t=1}^{T} \mathbb{E}_{\boldsymbol{p}_{\mathcal{T}'}} \left[ f_i(\boldsymbol{\theta}_t) - \min_{\boldsymbol{\theta}} \mathbb{E}_{\boldsymbol{p}_{\mathcal{T}'}} [f_i(\boldsymbol{\theta})] \right]$$

$$= \sum_{t=1}^{T} \mathbb{E}_{\boldsymbol{p}_{1:t}} \left[ w_{i_t} f_i(\boldsymbol{\theta}_t) - \min_{\boldsymbol{\theta}} \mathbb{E}_{\boldsymbol{p}_{\mathcal{T}'}} [f_i(\boldsymbol{\theta})] \right]$$

To test the generalization properties of the DASGRAD algorithm empirically, we unbalanced the MNIST training data set by reducing ninety percent the observations from the 1 and 3 digit. We set the importance weights to $w_{i_t} = (|L_i|/m)/p_{i_t}$, where $|L_i|$ is the count of the label $L$ associated with index $i$ in test over $m$, the number of test samples. As we see in Figure 3 using DASGRAD with the correct importance weights has the desired generalization properties when facing a domain shift.

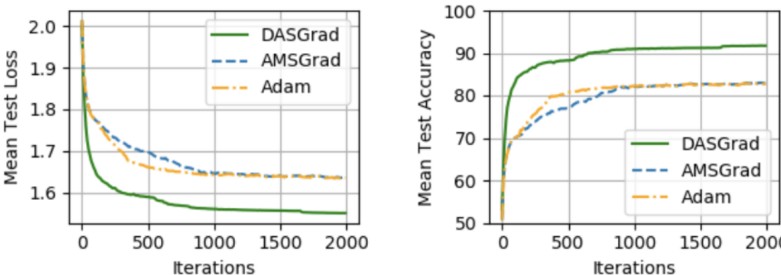

Figure 3: Trajectories of 20 random seeds for 2,000 iterations in convex optimization settings. Multiclass logistic regression on unbalanced MNIST dataset. We show the mean over 20 trajectories of training loss (left), training accuracy (center), and the improvement in accuracy of DASGrad with respect to AMSGRAD and ADAM with a 95% confidence interval (right)

## 7 CONCLUSION

Capability of learning from data efficiently is a prerequisite for practical success of complex learning models across various problem settings and application contexts. We have shown how double adaptive stochastic gradient descent methods enable efficient learning in a generalizable manner, while ensuring convergence improvement. We observed that DASGRAD algorithm outperforms currently prevalent variants of adaptive moment algorithms such as ADAM and AMSGRAD overall, in the context of the number of iterations required to achieve comparable performance, under the theoretical convergence guarantees in a stochastic convex optimization setting. With empirical validation in convex and non convex settings, we have shown that the advantages of DASGrad become more prominent with the increasing complexity of data and models, and with more variance in the gradients. We have also broadened our results to demonstrate generalization properties of our approach and its extensions to transfer learning, as well as intuitive connections to other learning scenarios.

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

# A  APPENDIX

## A.1  PROOF OF THEOREM 2

The proof of Theorem 2 assumes a convex differentiable objective function $f$, bounded diameter for the parameters, and bounded norm of the gradients for any trajectory of probabilities $\boldsymbol{p}_t \in \Delta_+^n$.

*Proof.* We build an upper bound of the expected regret using the convexity of the loss:
$$f(\boldsymbol{\theta}_t) - f(\boldsymbol{\theta}^*) \leq \langle \boldsymbol{g}_t, \boldsymbol{\theta}_t - \boldsymbol{\theta}^* \rangle = \mathbb{E}_n \left[ \langle \boldsymbol{g}_{i_t}, \boldsymbol{\theta}_t - \boldsymbol{\theta}^* \rangle \right]$$

While using DASGRAD the update of the parameter will be given by the stochastic update dependent on the training example $i_t$ and the current parameter $\boldsymbol{\theta}_t$:
$$\boldsymbol{\theta}_{t+1} = \Pi_{\Theta, \hat{\boldsymbol{V}}_{i_t}^{1/2}}(\hat{\boldsymbol{\theta}}_{t+1}) = \Pi_{\Theta, \hat{\boldsymbol{V}}_{i_t}^{1/2}}(\boldsymbol{\theta}_t - \alpha_t w_{i_t} \hat{\boldsymbol{V}}_{i_t}^{-1/2} \boldsymbol{m}_{i_t}) = \arg\min_{\boldsymbol{\theta} \in \Theta} || \hat{\boldsymbol{V}}_{i_t}^{1/4} (\boldsymbol{\theta}_t - \alpha_t w_{i_t} \hat{\boldsymbol{V}}_{i_t}^{-1/2} \boldsymbol{m}_{i_t}) ||$$

To bound the expected regret of the algorithm, we use the fact that:
$$\hat{\boldsymbol{\theta}}_{t+1} - \boldsymbol{\theta}^* = (\boldsymbol{\theta}_t - \boldsymbol{\theta}^*) - \alpha_t w_{i_t} \boldsymbol{m}_{i_t} / \sqrt{\hat{\boldsymbol{V}}_{i_t}}$$
$$|| \hat{\boldsymbol{V}}_{i_t}^{1/4}(\hat{\boldsymbol{\theta}}_{t+1} - \boldsymbol{\theta}^*) ||^2 = || \hat{\boldsymbol{V}}_{i_t}^{1/4}(\boldsymbol{\theta}_t - \boldsymbol{\theta}^*) ||^2 - 2\alpha_t w_{i_t} \langle \boldsymbol{m}_{i_t}, \boldsymbol{\theta}_t - \boldsymbol{\theta}^* \rangle + \alpha_t^2 w_{i_t}^2 || \hat{\boldsymbol{V}}_{i_t}^{-1/4} \boldsymbol{m}_{i_t} ||^2$$
$$= || \hat{\boldsymbol{V}}_{i_t}^{1/4}(\boldsymbol{\theta}_t - \boldsymbol{\theta}^*) ||^2 - 2\alpha_t w_{i_t} \langle \beta_{1_t} \boldsymbol{m}_{i_{t-1}} + (1 - \beta_{1_t}) \boldsymbol{g}_{i_t}, \boldsymbol{\theta}_t - \boldsymbol{\theta}^* \rangle + \alpha_t^2 w_{i_t}^2 || \hat{\boldsymbol{V}}_{i_t}^{-1/4} \boldsymbol{m}_{i_t} ||^2$$

We identify the first three components as the potential, the immediate cost, now with extra terms associated to the moving average, and the error.

**Lemma 1.** *For any $M \in S_+^d$ and convex feasible set $\Theta \subseteq \mathbb{R}^d$ with the projection operator $\Pi_{\Theta, M}$ let $u_1 = \Pi_{\Theta, M}(z_1)$ and $u_2 = \Pi_{\Theta, M}(z_2)$ then:*
$$|| \boldsymbol{M}^{1/2}(\boldsymbol{u}_1 - \boldsymbol{u}_2) || \leq || \boldsymbol{M}^{1/2}(\boldsymbol{z}_1 - \boldsymbol{z}_2) ||$$

Taking the expectation at time $t$, and using the extended norm reduction property of the projections from Lemma 1 we obtain the following inequality:
$$\mathbb{E}_{\boldsymbol{p}_t} \left[ || \hat{\boldsymbol{V}}_{i_t}^{1/4}(\boldsymbol{\theta}_{t+1} - \boldsymbol{\theta}) ||^2 \,\Big|\, \boldsymbol{\theta}_t \right] \leq \mathbb{E}_{\boldsymbol{p}_t} \left[ || \hat{\boldsymbol{V}}_{i_t}^{1/4}(\boldsymbol{\theta}_t - \boldsymbol{\theta}^*) ||^2 \,\Big|\, \boldsymbol{\theta}_t \right]$$
$$- \mathbb{E}_{\boldsymbol{p}_t} \left[ 2\alpha_t w_{i_t} \langle \beta_{1_t} \boldsymbol{m}_{i_{t-1}} + (1 - \beta_{1_t}) \boldsymbol{g}_{i_t}, \boldsymbol{\theta}_t - \boldsymbol{\theta}^* \rangle \,\Big|\, \boldsymbol{\theta}_t \right]$$
$$+ \alpha_t^2 \mathbb{E}_{\boldsymbol{p}_t} \left[ w_{i_t}^2 || \hat{\boldsymbol{V}}_{i_t}^{1/4} \boldsymbol{m}_{i_t} ||^2 \,\Big|\, \boldsymbol{\theta}_t \right]$$

Since $w_t$ is such that the interior product will be unbiased, then:
$$\mathbb{E}_{\boldsymbol{p}_t} \left[ || \hat{\boldsymbol{V}}_{i_t}^{1/4}(\boldsymbol{\theta}_{t+1} - \boldsymbol{\theta}) ||^2 \,\Big|\, \boldsymbol{\theta}_t \right] \leq \mathbb{E}_{\boldsymbol{p}_t} \left[ || \hat{\boldsymbol{V}}_{i_t}^{1/4}(\boldsymbol{\theta}_t - \boldsymbol{\theta}^*) ||^2 \,\Big|\, \boldsymbol{\theta}_t \right]$$
$$- 2\alpha_t \langle \beta_{1_t} \boldsymbol{m}_{t-1} + (1 - \beta_{1_t}) \boldsymbol{g}_t, \boldsymbol{\theta}_t - \boldsymbol{\theta}^* \rangle$$
$$+ \alpha_t^2 \mathbb{E}_{\boldsymbol{p}_t} \left[ w_{i_t}^2 || \hat{\boldsymbol{V}}_{i_t}^{1/4} \boldsymbol{m}_{i_t} ||^2 \,\Big|\, \boldsymbol{\theta}_t \right]$$

Finally rearranging the terms, summing until time $T$ and taking expectations:
$$R(\text{DASGRAD})_T \leq \sum_{t=1}^{T} \frac{1}{2\alpha_t(1 - \beta_{1_t})} \mathbb{E}_{\boldsymbol{p}_{1:t}} \left[ || \hat{\boldsymbol{V}}_{i_t}^{1/4}(\boldsymbol{\theta}_t - \boldsymbol{\theta}^*) ||^2 - || \hat{\boldsymbol{V}}_{i_t}^{1/4}(\boldsymbol{\theta}_{t+1} - \boldsymbol{\theta}^*) ||^2 \right]$$
$$+ \sum_{t=1}^{T} \frac{\alpha_t}{2(1 - \beta_{1_t})} \mathbb{E}_{\boldsymbol{p}_{1:t}} \left[ w_{i_t}^2 || \hat{\boldsymbol{V}}_{i_t}^{-1/4} \boldsymbol{m}_{i_t} ||^2 \right] \quad (1)$$
$$+ \sum_{t=1}^{T} \frac{\beta_{1_t}}{2(1 - \beta_{1_t})} \alpha_t || \hat{\boldsymbol{V}}_t^{-1/4} \boldsymbol{m}_{t-1} ||^2 + \sum_{t=1}^{T} \frac{\beta_{1_t}}{2\alpha_t(1 - \beta_{1t})} || \hat{\boldsymbol{V}}_t^{1/4}(\boldsymbol{\theta}_t - \boldsymbol{\theta}^*) ||^2$$

Last line is Cauchy-Schwarz and Young's inequality applied to the inner product of the extra terms associated with the moving average in the immediate cost.

$\square$

## A.2  PROOF OF COROLLARY 2.1

*Proof.* The proof of Corollary 2 is in the line of the improvements provided by Reddi et al. to the convergence proof of Kingma & Ba for ADAM, we adapt the arguments to the stochastic case. We assess separately each component of the expected regret from Equation 1.

Lemma 2 addresses the potential, Lemma 3 the error, and Lemma 6 and Lemma 4 the moving average terms. The proof of Corollary 2.1 is a consequence of all the previous Lemmas using the optimal probabilities while Corollary 1.1 is the case with uniform probabilities.

Following the sequence $\{\boldsymbol{\theta}_t\}_{t=1}^T$ of DASGRAD, with step size $\alpha_t = \alpha/\sqrt{t}$, averaging parameters $\beta_1 = \beta_{1_1}$ and $\beta_{1_t} \leq \beta_1$ for all $t \in [T]$ and $\gamma = \beta_1/\sqrt{\beta_2} < 1$. and bounded diameter $D$ for $\Theta$ and $||\nabla f_{i_t}(\boldsymbol{\theta})||_\infty \leq G$ for all $t \in [T]$ and $\boldsymbol{\theta} \in \Theta$.

**Lemma 2.** *From Equation 1 the potential component will satisfy:*

$$\sum_{t=1}^{T} \frac{1}{2\alpha_t(1-\beta_{1_t})} \mathbb{E}_{\boldsymbol{p}_{1:t}}\left[||\hat{\boldsymbol{V}}_{i_t}^{1/4}(\boldsymbol{\theta}_t - \boldsymbol{\theta}^*)||^2 - ||\hat{\boldsymbol{V}}_{i_t}^{1/4}(\boldsymbol{\theta}_{t+1} - \boldsymbol{\theta}^*)||^2\right] \leq \frac{D^2\sqrt{T}}{2\alpha(1-\beta_1)} \mathbb{E}_{\boldsymbol{p}_{1:T}}\left[||\hat{\boldsymbol{v}}_{i_T}^{1/4}||^2\right]$$

*Proof*
*One can decompose the potential in the following manner:*

$$\sum_{t=1}^{T} \frac{1}{2\alpha_t(1-\beta_{1_t})} \mathbb{E}_{\boldsymbol{p}_{1:t}}\left[||\hat{\boldsymbol{V}}_{i_t}^{1/4}(\boldsymbol{\theta}_t - \boldsymbol{\theta}^*)||^2 - ||\hat{\boldsymbol{V}}_{i_t}^{1/4}(\boldsymbol{\theta}_{t+1} - \boldsymbol{\theta}^*)||^2\right] \leq$$

$$\frac{1}{2\alpha_1(1-\beta_1)} \mathbb{E}_{\boldsymbol{p}_1}\left[||\hat{\boldsymbol{V}}_{i_1}^{1/4}(\boldsymbol{\theta}_1 - \boldsymbol{\theta}^*)||^2\right] - \frac{1}{2\alpha_T(1-\beta_1)} \mathbb{E}_{\boldsymbol{p}_{1:T}}\left[||\hat{\boldsymbol{V}}_{i_T}^{1/4}(\boldsymbol{\theta}_{T+1} - \boldsymbol{\theta}^*)||^2\right]$$

$$+\frac{1}{2(1-\beta_1)} \sum_{t=2}^{T} \left(\frac{1}{\alpha_t}\mathbb{E}_{\boldsymbol{p}_{1:t}}\left[||\hat{\boldsymbol{V}}_{i_t}^{1/4}(\boldsymbol{\theta}_t - \boldsymbol{\theta}^*)||^2\right] - \frac{1}{\alpha_{t-1}}\mathbb{E}_{\boldsymbol{p}_{1:t-1}}\left[||\hat{\boldsymbol{V}}_{i_{t-1}}^{1/4}(\boldsymbol{\theta}_{t-1} - \boldsymbol{\theta}^*)||^2\right]\right)$$

$$\leq \frac{1}{2\alpha_1(1-\beta_1)} \mathbb{E}_{\boldsymbol{p}_1}\left[||\hat{\boldsymbol{v}}_{i_1}^{1/4} \odot D \odot \boldsymbol{1}||^2\right]$$

$$+\frac{1}{2(1-\beta_1)} \sum_{t=2}^{T} \left(\frac{1}{\alpha_t}\mathbb{E}_{\boldsymbol{p}_{1:t}}\left[||\hat{\boldsymbol{v}}_{i_t}^{1/4} \odot D \odot \boldsymbol{1}||^2\right] - \frac{1}{\alpha_{t-1}}\mathbb{E}_{\boldsymbol{p}_{1:t-1}}\left[||\hat{\boldsymbol{v}}_{i_{t-1}}^{1/4} \odot D \odot \boldsymbol{1}||^2\right]\right)$$

$$\leq \frac{D^2}{2(1-\beta_1)} \left(\frac{1}{\alpha_1}\mathbb{E}_{\boldsymbol{p}_1}\left[||\hat{\boldsymbol{v}}_{i_1}||^2\right] + \sum_{t=2}^{T}\left(\frac{1}{\alpha_t} - \frac{1}{\alpha_{t-1}}\right)\mathbb{E}_{\boldsymbol{p}_t}\left[||\hat{\boldsymbol{v}}_{i_t}||^2\right]\right)$$

$$= \frac{D^2}{2\alpha_T(1-\beta_1)} \mathbb{E}_{\boldsymbol{p}_{1:T}}\left[||\hat{\boldsymbol{v}}_{i_T}^{1/4}||^2\right] = \frac{D^2\sqrt{T}}{2\alpha(1-\beta_1)} \mathbb{E}_{\boldsymbol{p}_{1:T}}\left[||\hat{\boldsymbol{v}}_{i_T}^{1/4}||^2\right]$$

*The first inequality comes from rearranging and the definition of $\beta_{1_t}$, the second inequality comes from the bounded diameter assumption applied to each entry of $\boldsymbol{\theta}_t - \boldsymbol{\theta}^*$ and using the Hadamard's product to represent the original matrix multiplication, the third inequality[5] comes from the definition of $\hat{\boldsymbol{v}}_{i_t} = max(\hat{\boldsymbol{v}}_{i_{t-1}}, \boldsymbol{v}_{i_t})$, the last equality comes from the property of the telescopic sequence. This completes the proof of Lemma 2.*

---

[5]The third inequality is of particular importance since Reddi et al. showed that it is one of the main flaws in the convergence analysis of ADAM and RMSPROP, and provided a simple fix to the adaptive moment methods that guarantees the non increasing property needed to achieve the telescopic sequence upper bound.

**Lemma 3.** *From Equation 1 the error component, once evaluated in the optimal probabilities $\hat{p}_t$ will satisfy:*

$$\mathbb{E}_{\hat{p}_t}\left[\hat{w}_{i_t}^2||\hat{V}_{i_t}^{1/4}m_{i_t}||^2\,\Big|\,\theta_t\right] = \mathbb{E}_n\left[||\hat{V}_{i_t}^{1/4}m_{i_t}||^2\,\Big|\,\theta_t\right] - Var_n\left(||\hat{V}_{i_t}^{1/4}m_{i_t}||\right)$$

*Proof*
*Creating a lower bound with Cauchy-Schwarz and showing that it is achievable with the optimal probabilities $\hat{p}_{i_t} \propto ||\hat{V}_{i_t}^{1/4}m_{i_t}||$.*

**Lemma 4.** *The first component of the extra terms associated with the moving average in Equation 1 will satisfy:*

$$\sum_{t=1}^{T}\frac{\alpha_t\beta_{1_t}}{2\alpha_t(1-\beta_{1_t})}||\hat{V}_t^{-1/4}m_{t-1}||^2 \leq \frac{\alpha Gd}{2\alpha(1-\beta_1)^3\sqrt{(1-\beta_2)(1-\gamma)}}$$

*Proof*
*Following very similar arguments as those from Lemma 6, we can get:*

$$\sum_{t=1}^{T}\frac{\alpha_t\beta_{1_t}}{2\alpha_t(1-\beta_{1_t})}||\hat{V}_t^{-1/4}m_{t-1}||^2 \leq \frac{\alpha}{2(1-\beta_1)^2\sqrt{(1\beta_2)(1-\gamma)}}\sum_{t=1}^{T}\beta_1^{T-t}||g_t||_1$$

$$\leq \frac{\alpha}{2(1-\beta_1)^2\sqrt{(1\beta_2)(1-\gamma)}}\sum_{t=1}^{T}\beta_1^{T-t}||G\odot\mathbf{1}||_1 \leq \frac{\alpha Gd}{2\alpha(1-\beta_1)^3\sqrt{(1-\beta_2)(1-\gamma)}}$$

*This completes the proof of Lemma 4.*

**Lemma 5.** *To finish the second component of the extra terms associated with the moving average in Equation 1, will satisfy:*

$$\sum_{t=1}^{T}\frac{\beta_{1_t}}{2\alpha_t(1-\beta_{1_t})}||\hat{V}_t^{1/4}(\theta_t-\theta^*)||^2 \leq \frac{D^2}{2\alpha(1-\beta_1)}\sum_{t=1}^{T}\sqrt{t}\beta_1^{T-t}||\hat{v}_t^{1/4}||^2$$

*Proof*

$$\sum_{t=1}^{T}\frac{\beta_{1_t}}{2\alpha_t(1-\beta_{1_t})}||\hat{V}_t^{1/4}(\theta_t-\theta^*)||^2 \leq \sum_{t=1}^{T}\frac{\beta_{1_t}}{2\alpha_t(1-\beta_{1_t})}||\hat{v}_t^{1/4}\odot D\odot\mathbf{1}||^2$$

$$= \frac{D^2}{2\alpha}\sum_{t=1}^{T}\sqrt{t}\frac{\beta_{1_t}}{(1-\beta_{1_t})}||\hat{v}_t^{1/4}||^2$$

$$\leq \frac{D^2}{2\alpha(1-\beta_1)}\sum_{t=1}^{T}\sqrt{t}\beta_1^{T-t}||\hat{v}_t^{1/4}||^2$$

*This completes the proof of Lemma 5.*

**Lemma 6.** *From Equation 1 the error component, once evaluated in the optimal probabilities $\hat{p}_t$, and the total error will satisfy that:*

$$\sum_{t=1}^{T} \frac{\alpha_t}{2(1-\beta_{1_t})} \mathbb{E}_{\hat{\boldsymbol{p}}_t} \left[ \hat{w}_{i_t}^2 ||\hat{\boldsymbol{V}}_{i_t}^{1/4}\boldsymbol{m}_{i_t}||^2 \,\Big|\, \boldsymbol{\theta}_t \right] \leq$$

$$\frac{\alpha\sqrt{1+\log(T)}}{2(1-\beta_1)^2\sqrt{(1-\beta_2)}(1-\gamma)} \sum_{h=1}^{d} || \, |\bar{\boldsymbol{g}}| \,|_{1:T,h} \, || - \sum_{t=1}^{T} Var_n \left( ||\hat{\boldsymbol{V}}_{i_t}^{1/4}\boldsymbol{m}_{i_t}|| \right)$$

*Proof*
*For Lemma 6 we follow Kingma & Ba, for every element at time $t$ of the error component:*

$$\frac{\alpha_t}{2(1\text{-}\beta_{1_t})} \mathbb{E}_n \left[ ||\hat{\boldsymbol{V}}_{i_t}^{-1/4}\boldsymbol{m}_{i_t}||^2 \right] \leq \frac{\alpha_t}{2(1\text{-}\beta_{1_t})} \mathbb{E}_n \left[ ||\boldsymbol{V}_{i_t}^{-1/4}\boldsymbol{m}_{i_t}||^2 \right]$$

$$= \frac{\alpha_t}{2(1\text{-}\beta_{1_t})} \mathbb{E}_n \left[ \frac{||\Sigma_{\tau=1}^{t}\beta_1(t)_\tau \boldsymbol{g}_{i_\tau}||^2}{\sqrt{v_{i_t}}} \right]$$

$$= \frac{\alpha_t}{2(1\text{-}\beta_{1_t})} \mathbb{E}_n \left[ \frac{||\Sigma_{\tau=1}^{t}\beta_1(t)_\tau^{1/2}\beta_1(t)_\tau^{1/2} \boldsymbol{g}_{i_\tau}||^2}{\sqrt{v_{i_\tau}}} \right]$$

$$\leq \frac{\alpha_t}{2(1\text{-}\beta_{1_t})} \mathbb{E}_n \left[ \left( \sum_{\tau=1}^{t} \beta_1(t)_\tau \right) \left( \sum_{\tau=1}^{t} \beta_1(t)_\tau \frac{||\boldsymbol{g}_{i_\tau}||^2}{\sqrt{v_{i_\tau}}} \right) \right]$$

$$\leq \frac{\alpha_t}{2(1-\beta_1)} \mathbb{E}_n \left[ \left( \sum_{\tau=1}^{t} \beta_1^{t-\tau} \right) \left( \sum_{\tau=1}^{t} \beta_1^{t-\tau} \frac{||\boldsymbol{g}_{i_\tau}||^2}{\sqrt{v_{i_\tau}}} \right) \right]$$

$$\leq \frac{\alpha_t}{2(1-\beta_1)^2} \mathbb{E}_n \left[ \left( \sum_{\tau=1}^{t} \beta_1^{t-\tau} \frac{||\boldsymbol{g}_{i_\tau}||^2}{\sqrt{(1-\beta_2)\Sigma_{\tau=1}^{t}\beta_2^{t-\tau}\boldsymbol{g}_{i_\tau}^2}} \right) \right]$$

$$\leq \frac{\alpha_t}{2(1-\beta_1)^2\sqrt{(1-\beta_2)}} \mathbb{E}_n \left[ \left( \sum_{\tau=1}^{t} \frac{\beta_1^{t-\tau}}{\sqrt{\beta_2^{t-\tau}}} \frac{||\boldsymbol{g}_{i_\tau}||^2}{|\boldsymbol{g}_{i_\tau}|} \right) \right]$$

$$= \frac{\alpha}{2(1-\beta_1)^2\sqrt{(1-\beta_2)}} \mathbb{E}_n \left[ \frac{1}{\sqrt{t}} \left( \sum_{\tau=1}^{t} \gamma^{t-\tau}||\boldsymbol{g}_{i_\tau}||_1 \right) \right]$$

*The first inequality follows the definition of the auxiliary vectors $\hat{\boldsymbol{v}}_t = max(\hat{\boldsymbol{v}}_{t-1}, v_t)$, the second inequality comes from the non negativity of $\beta_1(t)_\tau$. The third and fourth inequality comes from the decreasing property $\beta_1 \leq \beta_{1_1}$ and $\beta_{1_t} \leq \beta_{1_1}$ and the property of the geometric sequence. The fifth inequality comes from the non negativity of $\beta_2$ and $||g_{i_\tau}||^2$, the last equality uses the definition of the step size. Using induction one can show that:*

$$\sum_{t=1}^{T} \frac{\alpha_t}{2(1-\beta_{1_t})} \mathbb{E}_n \left[ ||\hat{\boldsymbol{V}}_{i_t}^{1/4}\boldsymbol{m}_{i_t}||^2 \right] \leq \sum_{t=1}^{T} \frac{\alpha}{2(1-\beta_1)^2\sqrt{(1-\beta_2)}} \mathbb{E}_n \left[ \left( \sum_{\tau=t}^{T} \frac{\gamma^{t-\tau}}{\sqrt{\tau}}||\boldsymbol{g}_{i_\tau}||_1 \right) \right]$$

$$(2)$$

*Continuing the proof of Lemma 6, let $k = \frac{\alpha}{2(1-\beta_1)^2\sqrt{(1-\beta_2)}}$, from Equation 2 we have that:*

$$\sum_{t=1}^{T} \frac{\alpha_t}{2(1-\beta_{1_t})} \mathbb{E}_n \left[ ||\boldsymbol{V}_{i_t}^{1/4}\boldsymbol{m}_{i_t}||^2 \right] \leq \sum_{t=1}^{T} k \, \mathbb{E}_n \left[ \frac{1}{\sqrt{t}} \left( \sum_{\tau=t}^{T} \gamma^{t-\tau} ||\boldsymbol{g}_{i_\tau}||_1 \right) \right]$$

$$= k \left( \sum_{t=1}^{T} \sum_{i_t=1}^{n} \frac{1}{n} ||\boldsymbol{g}_{i_t}||_1 \left( \sum_{\tau=t}^{T} \frac{\gamma^{t-\tau}}{\sqrt{\tau}} \right) \right)$$

$$\leq k \left( \sum_{t=1}^{T} \sum_{i_t=1}^{n} \frac{1}{n} ||\boldsymbol{g}_{i_t}||_1 \left( \sum_{\tau=t}^{T} \frac{\gamma^{t-\tau}}{\sqrt{t}} \right) \right)$$

$$= k \left( \sum_{t=1}^{T} \sum_{i_t=1}^{n} \frac{1}{n} ||\boldsymbol{g}_{i_t}||_1 \frac{1}{\sqrt{t}} \left( \sum_{\tau=t}^{T} \gamma^{t-\tau} \right) \right)$$

$$\leq \frac{k}{(1-\gamma)} \left( \sum_{t=1}^{T} \left( \sum_{i_t=1}^{n} \frac{1}{n} ||\boldsymbol{g}_{i_t}||_1 \right) \left( \frac{1}{\sqrt{t}} \right) \right)$$

$$= \frac{k}{(1-\gamma)} \sum_{h=1}^{d} \left( \sum_{t=1}^{T} \left( \sum_{i_t=1}^{n} \frac{1}{n} |\boldsymbol{g}_{i_t,h}| \right) \left( \frac{1}{\sqrt{t}} \right) \right)$$

$$\leq \frac{k}{(1-\gamma)} \sum_{h=1}^{d} \left( \sqrt{\sum_{t=1}^{T} \left( \sum_{i_t=1}^{n} \frac{1}{n} |\boldsymbol{g}_{i_t,h}| \right)^2} \sqrt{\sum_{t}^{T} \frac{1}{t}} \right)$$

$$\leq \frac{k\sqrt{1+\log(T)}}{(1-\gamma)} \sum_{h=1}^{d} \sqrt{\sum_{t=1}^{T} \left( \sum_{i_t=1}^{n} \frac{1}{n} |\boldsymbol{g}_{i_t,h}| \right)^2}$$

*The first equality comes from a change of indexes, the second inequality is an upper bound for the arithmetic sequence that begins at $t$, the third inequality is an upper bound for the geometric sequence, the fourth inequality comes is an application of Cauchy-Schwarz inequality, finally the fifth inequality is an upper bound for the arithmetic sequence.*

$$\sum_{t=1}^{T} \mathbb{E}_{\hat{\boldsymbol{p}}_t} \left[ \hat{w}_{i_t}^2 ||\hat{\boldsymbol{V}}_{i_t}^{1/4}\boldsymbol{m}_{i_t}||^2 \,\Big|\, \boldsymbol{\theta}_t \right] = \sum_{t=1}^{T} \mathbb{E}_n \left[ ||\hat{\boldsymbol{V}}_{i_t}^{1/4}\boldsymbol{m}_{i_t}||^2 \,\Big|\, \boldsymbol{\theta}_t \right] - \sum_{t=1}^{T} Var_n \left( ||\hat{\boldsymbol{V}}_{i_t}^{1/4}\boldsymbol{m}_{i_t}|| \right)$$

$$\leq \frac{\alpha\sqrt{1+\log(T)}}{2(1-\beta_1)^2\sqrt{(1-\beta_2)}(1-\gamma)} \sum_{h=1}^{d} ||| \bar{\boldsymbol{g}} |_{1:T,h} || - \sum_{t=1}^{T} Var_n \left( ||\hat{\boldsymbol{V}}_{i_t}^{1/4}\boldsymbol{m}_{i_t}|| \right)$$

*With the optimal probabilities $\hat{p}_{i_t}$, we complete the proof of Lemma 6.*

Finally we can combine the results from Lemma 2 to 5 and obtain the following bound for the expected regret of the general double adaptive algorithms:

**Corollary 2.1**

$$R(\text{DASGRAD}) \leq \frac{D^2 \sqrt{T}}{2\alpha(1-\beta_1)} \mathbb{E}_{\hat{\boldsymbol{p}}_{1:T}} \left[ ||\hat{\boldsymbol{v}}_{i_T}^{1/4}||^2 \right]$$

$$+ \frac{\alpha\sqrt{1+\log(T)}}{2(1-\beta_1)^2\sqrt{(1-\beta_2)}(1-\gamma)} \sum_{h=1}^{d} || \, |\bar{\boldsymbol{g}}|_{1:T,h} \, || - \sum_{t=1}^{T} \text{Var}_n \left( ||\hat{\boldsymbol{V}}_{i_t}^{1/4} \boldsymbol{m}_{i_t}|| \right)$$

$$+ \frac{\alpha Gd}{2\alpha(1-\beta_1)^3\sqrt{(1-\beta_2)}(1-\gamma)} + \frac{D^2}{2\alpha(1-\beta_1)} \sum_{t=1}^{T} \sqrt{t}\beta_1^{T-t}||\hat{\boldsymbol{v}}_t^{1/4}||^2$$

$$\square$$

