# OpenReview forum: "DASGrad: Double Adaptive Stochastic Gradient"
_ICLR.cc/2020/Conference — Reject_

### Official Review · AnonReviewer3 · 2019-10-22
**Official Blind Review #3**

**Rating:** 6

**Review:**

In this paper, the doubly adaptive stochastic gradient method (DASGrad) is introduced via augmenting adaptive moment methods with adaptive (as opposed to uniform) probabilities for data sampling. The convergence of the proposed method is analyzed in terms of regret bound and is compared to similar results for ADAM. The method is validated with experiments on both convex and non-convex objectives as well as in applications to transfer learning.

The paper is well written and provides novel, to the best of my knowledge, theoretical results on the convergence of adaptive moment methods when augmented with non-uniform data sampling, which is quite interesting. However, my main concern is the proposed implementation of non-uniform sampling. In particular, the adaptive probability of every training sample is set proportional to its gradient and, in theory, this has to be updated at every iteration. This essentially means that the full batch gradient is available at every iteration and, since the full gradient is available, wouldn’t it be better to use full batch gradient descent instead?

The proofs in the appendix repeat the known results in detail, while the definitions and proofs of the concepts related to the main contribution of this paper are rather short. In particular:
1. The expectations are not formally defined. The expectation with respect to p_{1:T} must be with respect to the join of all distributions from every iteration (?); sometimes there is conditioning on parameters and sometimes it disappears (proof of Lemma 1). This should be clarified.
2. It would also be helpful to see more detailed proof of Lemma 3 since it is not obvious and is the key ingredient of the presented results.
3. As a sanity check, I would expect that the results of Th. 1 and Th. 2 (or Corollary 1.1 and Corollary 2.1) should coincide when p is set to the uniform distribution. Is this correct and, if yes, can one show this?

Less important questions:
1. In Algorithm 2, there is a parameter J which controls the frequency of computation of optimal probabilities. How is this parameter set and what are the tradeoffs? After the parameters are updated, does it actually help to use such probabilities from the previous step when the parameters are in fact different?
2. Why parameter epsilon is introduced in Algorithm 2? How is it set?
3. Gradient aggregation methods are mostly criticized for their linear runtime in the number of examples n (see, e.g., Bottou et. al. 2016), while adaptive moment methods are widely used in practice since their runtime doesn’t depend on n. Since the proposed method is also linear in n, it would be interesting to see how it compares to aggregation methods in practice.
4. Conceptually, what is the difference of the results in Theorem 1 and Corollary 1.1 from the previous results in Kingma & Ba (2014) and Reddi et. al. (2018)?
5. In Lemma 4, there is unnecessary \alpha_t in the denominator. Does it change the final result?
6. To improve the runtime of the proposed algorithm, can one develop any approximation of optimal probabilities that wouldn’t require full data pass?

Minor:
1. p. 3, “Duchi et. al.” is this a reference?
2. functions phi and psi are called proximal; are they related to proximal operators?
3. typos: p. 3, “we first we”; p. 5, “an may”
4. p. 2, it is AMSGrad, not ADAM; betas aren’t defined
5. Inconsistent numbering: Theorems 1 and 2, but Corollaries 1.1 and 1.2


**Experience Assessment:**

I have read many papers in this area.

**Review Assessment: Checking Correctness Of Derivations And Theory:**

I carefully checked the derivations and theory.

**Review Assessment: Checking Correctness Of Experiments:**

I assessed the sensibility of the experiments.

**Review Assessment: Thoroughness In Paper Reading:**

I read the paper thoroughly.

---

### Official Review · AnonReviewer4 · 2019-10-23
**Official Blind Review #4**

**Rating:** 3

**Review:**

Summary:

This paper explores the combination of adaptive moment methods and adaptive data sampling procedures for stochastic optimization. The paper presents standard analysis in the stochastic convex setting which raised some question for me (presented below). The empirical results are promising but in my opinion can be further improved through minor changes to experimental procedure.


Overall:

1) There are a few instances of undefined notation. For example, $\beta_1(t)$ in the definition of Adam/AMSGrad or $V_t$ (the latter I assume is the average over all datapoints?).

2) Theorem 1 is stated very informally with some notation left unexplained (e.g. V_t above or E_n[...]). Further, it is not clear to me whether the authors are proving results in the online convex optimization setting or the stochastic convex optimization setting (though, there is some equivalence between them). Finally, I am concerned by the presentation of Theorem 1 and Corollary 1.1 as it has been shown by Reddi et al. that Adam does not have bounded regret in general without algorithmic changes. In particular, bimodal gradient noise distributions can be designed such that the average Regret is unbounded (see Reddi et al. for details). While these issues can be addressed through modifications to the standard Adam algorithm I am unable to find a suitable discussion within the paper.

3) Theorem 2 introduces DASGrad which has not yet been clearly defined (Algorithm 1 depends on averaging functions of the past gradients for flexibility --- which averaging functions are being used here?). Once again, the notation Var_n appears whose definition is not clear to me. Additionally, could you provide a citation for the optimal adaptive probabilities and clearly state under which conditions these adaptive probabilities are optimal?

The crux of Theorem 2 depends on the informal statements present in Section 3.3, in which it is claimed that the variance of the second order moments are sufficiently large to dominate the sum of gradient histories. I am having a hard time seeing how this would be true generally and I believe that further theoretical analysis is necessary to justify this discussion.

4) The introduction of Algorithm 2 felt like a bait-and-switch. This algorithm seems intuitively sensible and uses the same check-pointing ideas that have been introduced in recent variance reduction methods (e.g. [1]). However, this is a different algorithm which is not analyzed in this paper.

Additionally, the fourth line of Algorithm 2 says to compute p_t lying in the simplex --- I believe this is a typo? Further, could you explain the introduction of epsilon in line 4. This seems unnecessary given the propto symbol but I am worried I am missing something.

5) The empirical results introduce an additional deviation from the proposed algorithm by using batch sizes of 32. Could the authors please explain how the adaptive sampling probabilities are used in this setting? (My apologies if this is obvious, I am not overly familiar with such methods).

Overall I felt that the empirical results were fairly promising but some questions remained for me. The toy experiments in section 6.1 were illustrative but seem to only consider adaptive probabilities without the adaptive moments?

Section 6.3 was also an interesting experimental setup which is well suited to DASGrad but I felt that the baseline comparison was unfair. DASGrad was provided with the ground-truth balance statistics in the importance weights while the baselines were trained without any correction. I would recommend that the authors explore combinations of enabling balance correction in the baseline methods (possibly through reweighting the loss according to the balance statistics) or requiring DASGrad to compute the optimal probabilities through the adaptive moments as suggested throughout the rest of the paper.

Minor:

- Pg 3, typo "we first we extend"


References

[1] Accelerating Stochastic Gradient Descent using Predictive Variance Reduction, Rie Johnson and Tong Zhang


**Experience Assessment:**

I have read many papers in this area.

**Review Assessment: Checking Correctness Of Derivations And Theory:**

I assessed the sensibility of the derivations and theory.

**Review Assessment: Checking Correctness Of Experiments:**

I carefully checked the experiments.

**Review Assessment: Thoroughness In Paper Reading:**

I read the paper at least twice and used my best judgement in assessing the paper.

---

> ### Author Response · Authors · 2019-11-15
> **On the Convergence of Adam, AMSGrad and DASGrad.**
>
> 1) Thanks for this comment, $\beta_{1}(t)$ should be $\beta_{1}{t}$ from adaptive moment methods.
>
> 2) The matrix $V_{t}$ is defined in page 5 on the algorithm. Thanks for pointing your concerns regarding the convergence of Adam, we used AMSGrad fixed version and proof to account for the Adam convergence problems we explain it in the Corollary but we can make it explicit in the main paper. We built DASGrad on top of the AMSGrad algorithm as one can see in the convergence proof in the Corollary.
>
> 3) The $Var_{n}$ stands for empirical variance, we can clarify in the main paper, it is carefully explained in the Corollary. "could you provide a citation for the optimal adaptive probabilities", this is the main result of our paper explicitly showing the optimal adaptive probabilities for the method and we show it is a consequence of Cauchy-Schwarz.
>
> 4) Thanks for the comment, we are happy to see that recently variance reduction methods are gaining some interest in deep learning literature. The epsilon in the algorithm is included to avoid numerical problems, we can clarify in the footnote.
>
> 5) Regarding the batch size, we restricted our proof for adaptive probabilities for the mini batch setting, the interaction between batch size and adaptive probabilities is an interesting problem to be investigated in the future.

---

### Official Review · AnonReviewer1 · 2019-10-23
**Official Blind Review #1**

**Rating:** 3

**Review:**

In the paper, the authors propose a general form that covers most stochastic gradient methods so far, e.g. stochastic gradient descent, Adam,  or adaptive probabilities methods. Then, they also provide a convergence analysis of convex problems.  In the experiments, they compared the proposed DASGRAD method with Adam, AMSGrad or SGD. Experimental results show that the proposed method converges faster than compared methods.  Following are my concerns:

1) It is important to cite properly. There are some related works missing in the paper. e.g.,  [1][2][3] for adaptive methods, and [4] adaptive sampling methods.
2) In corollary 1.1, the rightmost term is in the order of \sum_t=1^T \sqrt{t}, which should be O(T\sqrt(T)). If we divide both sides by T, the right side of the inequality is still upper bounded by \sqrt{T}, it is not convergent. Otherwise, please correct me and show the convergence rate of the proposed method.
3) The same question in Corollary 2.1 about the convergence result.
4) Some variables are not defined clearly in the paper, e.g., \bar g and  \beta_{1_{t}}  in corollary 1.1.
5) Algorithm 2 requires to compute p every J iterations. How about the time complexity of sampling? Please also show the convergence regarding time in the experiments.

[1] Zhou, Dongruo, et al. "On the convergence of adaptive gradient methods for nonconvex optimization." arXiv preprint arXiv:1808.05671 (2018).
[2] Chen, Xiangyi, et al. "On the convergence of a class of adam-type algorithms for non-convex optimization." arXiv preprint arXiv:1808.02941 (2018).
[3]Zaheer, Manzil, et al. "Adaptive methods for nonconvex optimization." Advances in Neural Information Processing Systems. 2018.
 [4]Csiba, Dominik, Zheng Qu, and Peter Richtárik. "Stochastic dual coordinate ascent with adaptive probabilities." International Conference on Machine Learning. 2015.



**Experience Assessment:**

I have published one or two papers in this area.

**Review Assessment: Checking Correctness Of Derivations And Theory:**

I assessed the sensibility of the derivations and theory.

**Review Assessment: Checking Correctness Of Experiments:**

I assessed the sensibility of the experiments.

**Review Assessment: Thoroughness In Paper Reading:**

I read the paper at least twice and used my best judgement in assessing the paper.

---

> ### Author Response · Authors · 2019-11-15
> **Response on the correctness of the convergence proof**
>
> 1) Thanks for pointing this missing literature. It is nice to know that there is some increasing interest in adaptive probabilities for non convex objectives. Regarding "Stochastic dual coordinate ascent with adaptive probabilities", there has been a lot of variants for classic convex settings we could not mentioned them all.
>
> 2) Thanks for the comment on the order of the right hand part of our convergence proof. The $O(T\sqrt(T))$ is indeed and upper bound for the term although is not tight, in fact $\sum_{t=1}^T \sqrt{t}$ is still in sub linear order. This term is the same as the term in AMSGrad and Adam like algorithms, in the convergence proof of those the term is implicit in the learning rate. The same is true for question 3.
>
> 4) The variable $\beta_{1,t}$ is the exponential smoothing term of Adam. We defined the variable g in page 15 in the Corollary, thanks for the observation. We will clarify
>
> 5) Thanks for this comment, the computational burden of calculating the optimal adaptive probabilities is difficult to address currently due to limitations of dynamic graphs that compute the average gradients by default. Because of this we only provided an empirical validation of the results against iterations, instead of time.

---

### Decision · Program_Chairs · 2019-12-19

**Decision:**

Reject

**Comment:**

The reviewers were confused by several elements of the paper, as mentioned in their reviews and, despite the authors' rebuttal, still have several areas of concerns.

I encourage you to read the reviews carefully and address the reviewers' concerns for a future submission.